

# Captive-rearing changes the gut microbiota of the bumblebee *Bombus lantschouensis* native to China

Feng Zhou*, Shuning Sun*, Xinge Song, Yuying Zhang, Zhuanxia Li and Jiani Chen

College of Science, The Northwest Normal University, Lanzhou, Gansu, China
* These authors contributed equally to this work.

## ABSTRACT

Bumblebees play crucial roles as pollinators in both natural agricultural and ecological systems. Their health and overall fitness are significantly affected by the gut microbiota, which can maintain intestinal homeostasis in hosts by regulating their nutritional metabolism. However, information about the diversity of the gut microbiota and related functional changes during artificial rearing of native species is currently limited. This study investigated the dynamic remodeling of gut microbiota in the Chinese native bumblebee *Bombus lantschouensis* under captive rearing, supported by 16S rRNA amplicon sequencing of bacterial DNA. The typical microbial community composition of the bumblebee was detected in the gut of wild *B. lantschouensis*, with species of genus *Gilliamella* and *Snodgrassella* identified as the dominant strains. Conversely, the microbiota of the captive-reared group showed increased diversity and decreased abundance of certain species of microorganisms. The populations of *Bifidobacterium*, *Saccharibacter*, and *Lactobacillus*, including Firm-4 and Firm-5, were dramatically increased after captive-rearing and became the dominant bacteria, while *Gilliamella* and *Snodgrassella* were strikingly reduced. Notably, this study found that pathogenic bacteria appeared in the intestines of wild-caught *B. lantschouensis* and disappeared when the host was reared under captive conditions. This study shows microbial community changes in bumblebees and facilitates the study of physiological metabolism in the commercial rearing of insects.

## INTRODUCTION

Pollinating insects play a pivotal role in ecosystems, serving as vital agents in plant reproduction, crop security, and human well-being (*McCallum, McDougall & Seymour, 2013*). They feed on the pollen and nectar of flowering plants and move among different flowers, which frequently leads to interactions with conspecific and heterospecific individuals, therefore establishing extensive microbial interactions through pollination (*Tylianakis et al., 2008*). Gut microbiota refers to the large number of microorganisms in the intestinal tract of pollinators (*Schoeler & Caesar, 2019*). As crucial global pollinators, bees serve as an important model for studying gut microbial communities (*Motta &*

Corresponding author
Feng Zhou, fengzhou@nwnu.edu.cn

*Moran, 2024*). Only nine species of bacteria are predominant in the bee gut, and these bacteria are spread through interactions with different bees individuals (*Kwong & Moran, 2016*). These bacteria assist hosts with a variety of physiological functions such as metabolism, nutrient uptake, and immune regulation (*Chen & Wang, 2022*). These bacteria help bees decompose and utilize pollen grains, degrade toxic substances in the environment, and stimulate the immune system to prevent the invasion of pathogenic microorganisms (*Wang et al., 2022*; *Haag et al., 2023*). These bacteria can even regulate their host's foraging preference for flowers by effecting the host's nutrient regulation, including sugar metabolism (*Fouks & Lattorff, 2011*). These bacteria evolve in specific populations and locations within the bee gut in order to maintain microenvironment homeostasis. Their functions are usually dynamic and respond to changes in species, age, or developmental stages (*Zhou et al., 2023*). These bacteria restructure the composition and proportion of their population in the gut under different environmental factors such as pH, food, or other exogenous substances (*Patterson et al., 2016*). Diet can also affect the function and community composition of the gut microbiota (*Schoeler & Caesar, 2019*). Therefore, investigating the structure and population dynamics of gut microorganisms is an effective way to study their role in pollination functions.

Bumblebees have irreplaceable roles in pollination due to their distinctive morphology and behavior (*Zhang & Zheng, 2022*). An increasing number of recent studies have focused on the gut bacteria of bumblebees because of their key role in the host's physiological metabolism, behavior, and pollination functions (*Newbold et al., 2015*). For example, *Roy et al. (2023)* found that the difference in pollination ability between captive-reared and wild-caught bumblebees may be due to differences in glycometabolism resulting from changes in the gut microbiomes of these bumblebees. With the increase in captive rearing and commercial pollination applications, it is becoming increasingly important to understand the role of the gut microbiome. Captive rearing provides an artificial environment programmed for bumblebee growth, generating new challenges in feeding, parasitic disease control, and breeding (*Roy et al., 2023*). Prior research has shown that when bee colonies were reared with artificial diets and controlled environments, the species and diversity of gut microorganisms significantly changed in some model bees such as *Bombus terrestris*. This provides a preconceived model for investigating the functions of bee gut microbiota (*Krams et al., 2022*). More research may be needed on non-model or native populations to examine what roles microbiota play in a healthy colony, pollination efficacy, and response to ecological factors for both wild bumblebee and captive-reared populations.

*B. lantschouensis* is a species of bumblebee native to China, mainly found in the northwest region of the country (*An et al., 2014*). This species is a candidate for commercial rearing and greenhouse pollination because of its excellent pollination performance and potential for commercial breeding (*Dong et al., 2020*). Six major bacterial lineages have been found in the gut of *B. lantschouensis*. It has also been found that the gut bacteria can affect reproduction, immune function, and adaptability in *B. lantschouensis*. For example, PGRP-LC (Bl-PGRP-LC) from *B. lantschouensis* can respond to the infection

of gram-negative *Escherichia coli* by binding directly to the DAP-type PGNs (*Liu et al., 2020*). Additionally, the proportion of *Bifidobacterium* in the gut responds to the reproductive activity of *B. lantschouensis* (*Wang et al., 2019*). However, based on changes in ecological and physiological factors, the dynamic response and roles of the bacteria in the gut of *B. lantschouensis* are very limited.

This study used the 16S amplicon sequencing method. This strategy was first applied to the phylogenetic analysis of bacterial classification (*Lane et al., 1985*) and has since been widely adopted to study the diversity of microbial communities in various animals. Different hypervariable regions of 16S rRNA have been evaluated and selected for amplicon sequencing in various populations (*Jones et al., 2022*), especially in environment-related species (*Feng & McLellan, 2019*). Using 16S amplicon sequencing, the present study investigated the dynamic patterns of gut microbiota in *B. lantschouensis* under captive rearing. The findings of this study can provide useful information for future studies on artificial feeding and pollination.

## MATERIALS AND METHODS

### Bumblebee sample collection

A total of 60 wild-caught workers of *B. lantschouensis* were collected from three sites on Xinglong Mountain (104°0′30.30″N, 35°47′0.92″E, 2,335.58 m) in Lanzhou city, Gansu Province in July 2022, with each site containing at least five individuals. After collection, the insects were identified according to morphological characters followed by cytochrome oxidase I (COI) gene sequencing (*Dai et al., 2024*). The primers for COI amplification were LepF1: ATTCAACCAATCATAAAGATAT, LepR1: TAAACTTCTGGATGTCCAAAAA (*Cha et al., 2007*). The identified bees were divided into two groups: wild-caught and captive-reared. The wild-caught group was immediately dissected to collect the intestines. The captive-reared group was given a solution consisting of 25% sucrose, 25% rapeseed pollen, and 50% sterile water, which was changed daily. This group was kept at 28 ± 1 °C and placed in an artificial climate box where the humidity was maintained at 55 ± 5%. The environment was kept dark and well-ventilated. The colony of 40 bees, with no queen, was generally active and healthy, while dead individuals were manually removed. After 20 days of feeding, intestinal samples were taken from the 30 healthy individuals with the same method used for the wild-caught group. All intestinal tissues were immediately frozen in liquid nitrogen and stored at −80 °C for further use.

### Extraction of the gut DNA

Gut samples from 30 bees of each treatment group were divided into six biological replicates including five gut samples. The total DNA was extracted using the Wizard Genomic DNA Purification Kit (Promega, Madison, WI, USA) following the manufacturer's protocol. The quality of the DNA extraction was evaluated using 1% agarose gel electrophoresis (*Gerasimidis et al., 2016*) and the concentration and purity of the DNA were determined using a NanoDrop 2000 UV-VIS spectrophotometer. The high-quality DNA samples were stored at −80 °C until further processing.

## Illumina sequencing and taxonomy classification

DNA samples with replicates in each group were used for polymerase chain reaction(PCR) amplification. The hypervariable V3–V4 region of the bacterial 16S rRNA gene was amplified with the forward primer F (5′-CCTACGGGNGGCWGCAG-3′) and the reverse primer R (5′-GACTACHVGGGTATCTAATCC-3′). The PCR reaction conditions were 95 °C for 5 min, 95 °C for 30 s, 50 °C for 30 s, and 72 °C for 1 min, with 25 cycles in a reaction volume of 10 μl, and the DNA polymerase used was 2xF8 FastLong PCR MasterMix. PCR products were detected by 2% agarose gel electrophoresis and purified using the QIAquick Gel Extraction Kit (Qiagen). After sample-specific index sequences were added, PCR products were further quantified and an amplicons library was constructed. Finally, paired-end sequencing was performed on the Illumina NovaSeq 6000 platform (Illumina, San Diego, CA, USA). After filtration, merging, and quality control, the amplicon sequence variant (ASV) classification was assigned using the Naïve Bayes (*Zhang, 2016*) algorithm implemented in the q2 feature classifier using the BEE database (version 1.0). The core of the ASV methodology lies in the bioinformatic processing of the quality-filtered reads, which is achieved through DADA2 and Deblur (*Callahan et al., 2016*). ASV-based approaches assign each unique sequence to its own variant and calculate the numbers of sequences for each classification level (*e.g.*, phylum and genus). Phylogenetic tree analysis was constructed by the MEGA XI (*Tamura, Stecher & Kumar, 2021*) using the neighbor-joining method with a bootstrap value of 1,000. It was visualized using iTOL.v6 (https://itol.embl.de/).

## Diversity analysis and functional prediction

To find the microbial community diversity within and between groups of captive-reared and wild-caught *B. lantschouensis*, the spliced raw tags were strictly filtered using FastQC to obtain high-quality tags (*Bokulich et al., 2013*). These were compared using the SILVA 138.1 database to detect chimeric sequences, which were then removed to obtain the final effective tags (*Edgar et al., 2011*). Alpha diversity parameters, including the Chao 1, Abundance-based Coverage Estimator (ACE), Shannon, and observed ASV indexes, were employed for community richness comparisons and visualized in R using the randomly extracted sequencing abundances per sample. Beta diversity was calculated and visualized by generating principal coordinate plots based on weighted and unweighted UniFrac distance metrics using the Quantitative Insights Into Microbial Ecology (QIIME2) pipeline (http://qiime.org/; *Edgar et al., 2011*). Phylogenetic Investigation of Communities by Reconstruction of Unobserved States (PICRUSt2) was used to predict functional profiles of microbials in the gut microbiota (*Douglas, 2015*). The abundance of Kyoto Encyclopedia of Genes and Genomes (KEGG) pathways and the MetaCyc database were used to test the differences of functional pathways between the two groups. The linear discriminant analysis (LDA) and linear discriminant analysis effect size (LEfSe; *Segata et al., 2011*; $p < 0.05$, $q < 0.1$, LDA > 2.0) were performed to identify potential microbial biomarkers between groups. This method uses the nonparametric Kruskal–Wallis test with default settings in a rank-sum test to identify biomarkers (*Scheirer, Ray & Hare, 1976*).

## Statistical analysis

Statistical tests were performed using the chi-squared test implemented in R for the 16S rRNA data. The data of the two processing groups were analyzed by an unpaired *t*-test to identify data differences (*Mishra et al., 2019*), while the data differences of multiple groups were determined by permutational multivariate analysis of variance (PERMANOVA; *Carson et al., 2009*). $P < 0.05$ was considered statistically significant. The extent of the differences between the samples and groups was analyzed by nonmetric multidimensional scaling (NMDS) analysis and Principal Coordinates Analysis (PCoA; *Huson & Weber, 2013*).

## RESULTS

### The statistics of amplicon sequencing

To analyze the bacterial communities of the entire gut, a total of 12 DNA samples from 10 were qualified (one DNA sample from the wild-caught groups and one from the captive-reared groups each failed). Qualified samples were amplified V3–V4 regions and generated 724,851 raw reads with an average of 80,110 reads for each sample. The average percentage of reads for filtration, merging, and chimeric removal were 91.2%, 90.6% and 84.8%, respectively (Table 1). After a series of quality controls, a total of 678,089 reads were produced for amplicon sequence variant (ASV) assembly for the analysis of gut microbial species. Following the removal of chimera and low abundance sequences, 153 different ASVs were identified according to a 97% pairwise nucleotide sequence threshold using QIIME2 software. The sequence length distribution showed that 125 ASVs (82%) demonstrated a length between 426–431 nucleotides (Fig. S1). The annotation of ASVs showed that 91.5% members were collocated under a clear bacterial taxonomy based on the BEE database (*Elsik et al., 2016*).

### The composition of gut microbiota in *B. lantschouensis*

To show the taxonomic evolution of gut microbiota for *B. lantschouensis*, a clear phylogenetic relationship of bacterial classification order was demonstrated in the clades of a phylogenetic tree, which was built using the top 100 ASVs with the highest abundance (Fig. 1A). Five clades were clustered in the phylogenetic tree and showed a clear phylogenetic relationship of bacteria at the phylum level. The composition of gut microbiota in each classification level for *B. lantschouensis* was analyzed. At the phylum level, *Proteobacteria* were most abundant with 72%, followed by *Firmicutes* (20.9%) and *Actinobacteria* (7.0%). At the class level, the richest microbiota composition was found to be *Gammaproteobacteria* (57.7%, belonging to Proteobacteria), followed by *Bacilli* (20.9%, belonging to Firmicutes), *Alphaproteobacterial* (14.3%, belonging to Proteobacteria), and *Actinobacteria* (7.0%). At the order level, the composition of *Orbales* (32.26%, belonging to Gammaproteobacteria), *Betaproteobacteriales* (25.32%, belonging to Gammaproteobacteria), and *Lactobacillales* (20.74%, belonging to Bacilli) was found to be the most abundant. At the family level, *Orbaceae bacterium* (belonging to Orbales) was found to be the most abundant. At the genus level, *Gilliamella* (32.25%, belonging to Orbaceae) and *Snodgrassella* (25.32%, belonging to Neisseriaceae) were found to be the

**Table 1 Summary of quality control in 16S rRNA amplicon sequencing of gut microbiota in *B. lantschouensis*.**

| Id | Group | Input | Filtered | Percentage of input passed filter | Denoised | Merged | Percentage of input merged | Non-chimeric | Percentage of input non-chimeric |
|----|-------|-------|----------|-----------------------------------|----------|--------|----------------------------|--------------|----------------------------------|
| 1 | Wild-caught | 83,586 | 76,803 | 91.89 | 76,647 | 76,352 | 91.35 | 69,110 | 82.68 |
| 2 | Wild-caught | 81,330 | 74,475 | 91.57 | 74,387 | 74,186 | 91.22 | 68,051 | 83.67 |
| 3 | Wild-caught | 84,237 | 75,050 | 89.09 | 74,935 | 74,706 | 88.69 | 69,649 | 82.68 |
| 4 | Wild-caught | 81,290 | 74,738 | 91.94 | 74,619 | 74,307 | 91.41 | 68,170 | 83.86 |
| 5 | Wild-caught | 77,502 | 70,531 | 91.01 | 70,403 | 70,099 | 90.45 | 66,138 | 85.34 |
| 6 | Captive-reared | 79,229 | 72,589 | 91.62 | 72,459 | 71,891 | 90.74 | 68,114 | 85.97 |
| 7 | Captive-reared | 74,431 | 69,147 | 92.9 | 69,022 | 68,883 | 92.55 | 66,244 | 89 |
| 8 | Captive-reared | 76,128 | 70,650 | 92.8 | 70,494 | 70,015 | 91.97 | 66,681 | 87.59 |
| 9 | Captive-reared | 76,018 | 70,852 | 93.2 | 70,608 | 70,007 | 92.09 | 66,512 | 87.5 |
| 10 | Captive-reared | 87,345 | 75,059 | 85.93 | 74,963 | 74,405 | 85.19 | 69,420 | 79.48 |

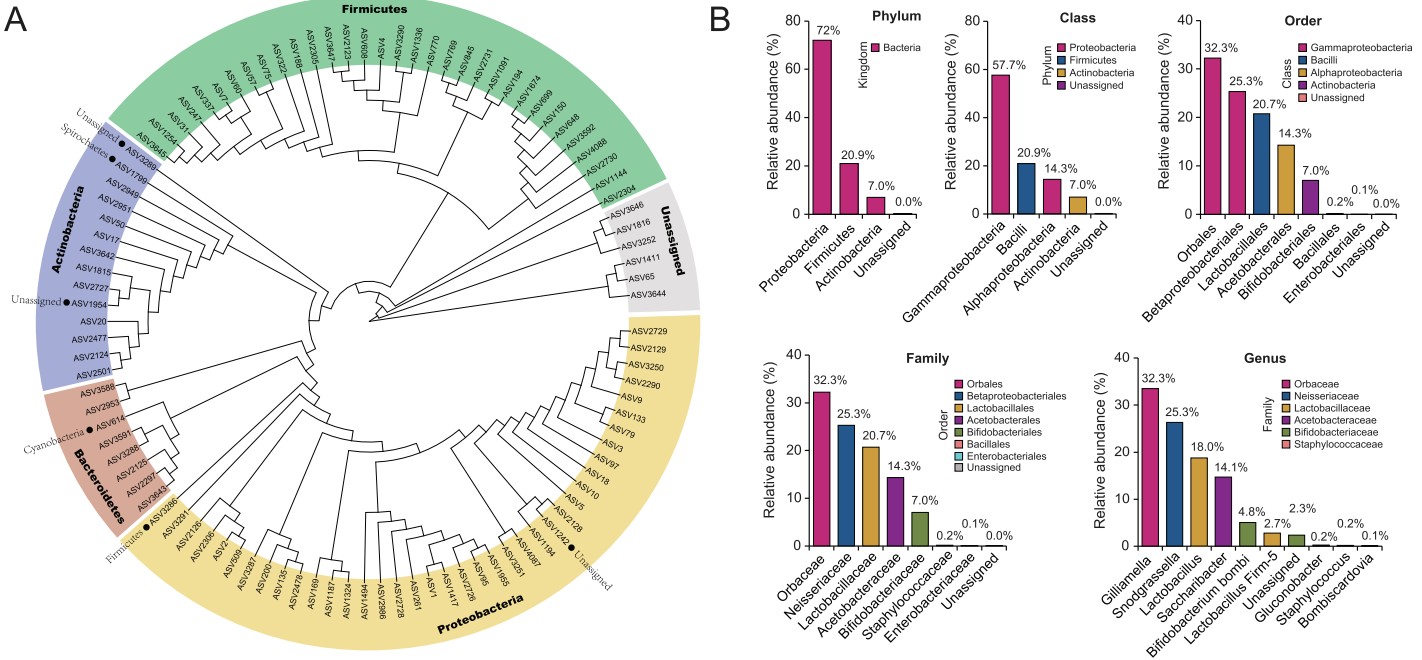

**Figure 1 The composition of gut microbiota in *B. lantschouensis*.** (A) The phylogenetic analysis of gut microbiota of *B. lantschouensis*. (B) The abundance of the main taxonomic categories of gut microbiota of *B. lantschouensis*.

most abundant, followed by *Lactobacillus* (18%, belonging to Lactobacillaceae), and *Saccharibacter* (14.1%, belonging to Acetobacteraceae; Fig. 1B).

## Diversity differences in the gut microbiota between wild—caught and captive-reared groups of *B. lantschouensis*

The ASVs were compared for the captive-reared and wild-caught groups. The two groups had 13 shared ASVs, whereas 53 ASVs were unique to the wild-caught group and 83 ASVs were unique to the captive-reared group (Fig. 2A). Rarefaction curves indicated that the

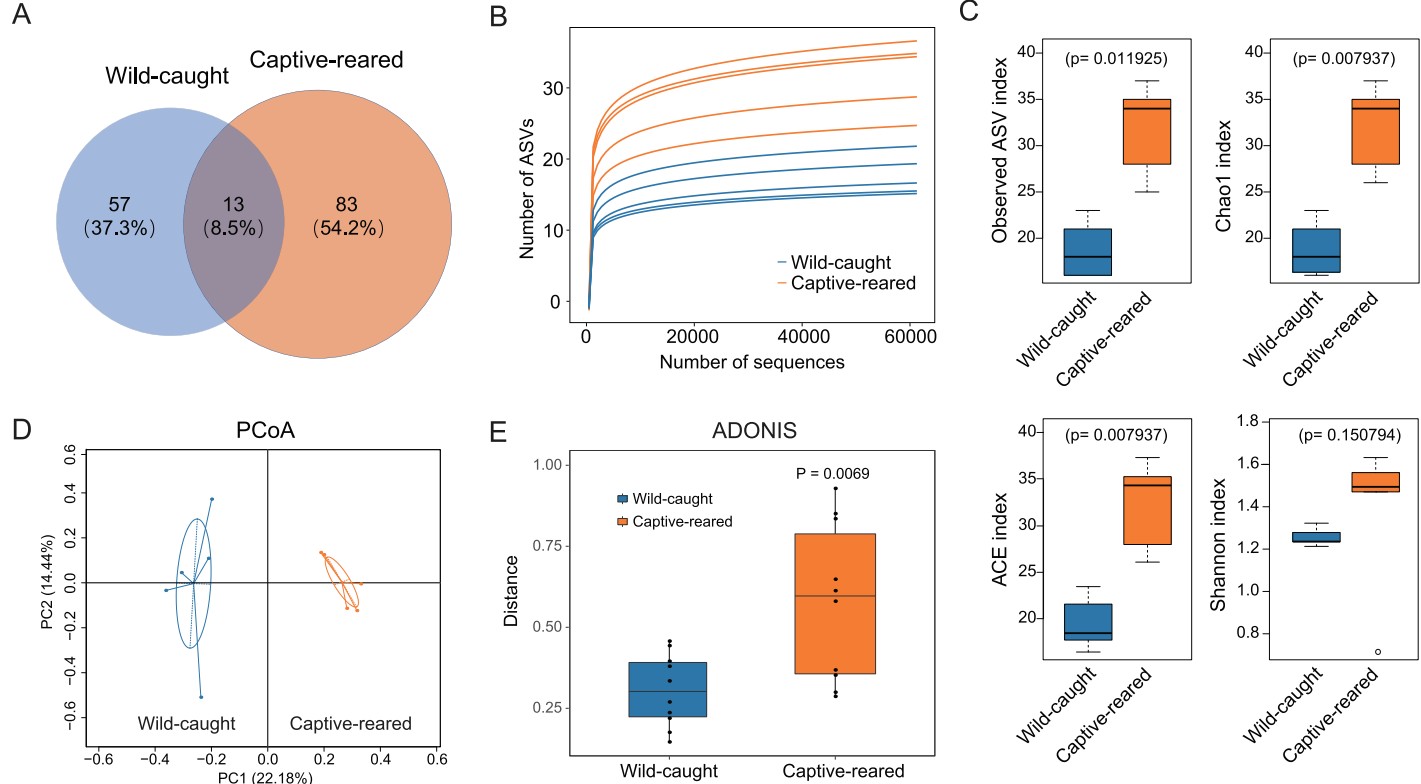

**Figure 2** **The diversity analysis of gut microbiota between wild-caught and captive-reared *B. lantschouensis*.** (A) Venn diagram of significant differences in ASV number of gut microbiota between wild-caught and captive-reared *B. lantschouensis*. (B) Rarefaction curves of each sample in the 16S rDNA analysis. In the rarefaction curves the horizontal axis represents the proportion of selected ASVs (%) and the vertical axis shows the groups' richness (number of ASVs). (C) Alpha diversity index analysis of gut microbiota between wild-caught and captive-reared *B. lantschouensis*. (D) UniFrac principal coordinates analysis (PCoA) of gut microbiota between wild-caught and captive-reared *B. lantschouensis*. Ellipses are 95% confidence areas. Group differences were tested by permutational multivariate analysis of variance (PERMANOVA) among all groups. (E) Adonis analysis of gut microbiota between wild-caught and captive-reared *B. lantschouensis*.

bacterial species richness and diversities stabilized as the number of sequences increased (Fig. 2B). The alpha diversity indices (observed ASVs, Chao1, ACE, Shannon) of the two groups were measured. The richness (Chao 1, ACE, and observed ASVs) of captive-reared bumblebee gut bacterial communities significantly increased ($P < 0.05$) compared to those in the wild-caught group (Fig. 2C). According to the principal coordinates analysis (PCoA) of weighted UniFrac, the gut microbiota of the captive-reared group showed a significant deviation from the wild-caught group, with a $P$ value of 0.006 (Fig. 2D). This finding was supported by an adonis analysis (PERMANOVA; Fig. 2E), verifying the difference of the gut microbiota between the two groups was significant.

## Effects of captive rearing on the gut microbiota composition in *B. lantschouensis*

To investigate the effect of captive rearing on the changes of the gut microbiota of wild *B. lantschouensis*, the gut bacterial community of captive-reared bumblebees were compared with the wild-caught group. At the phylum level, almost 100% of Proteobacteria were observed in the wild-caught group of *B. lantschouensis*, while the composition of the

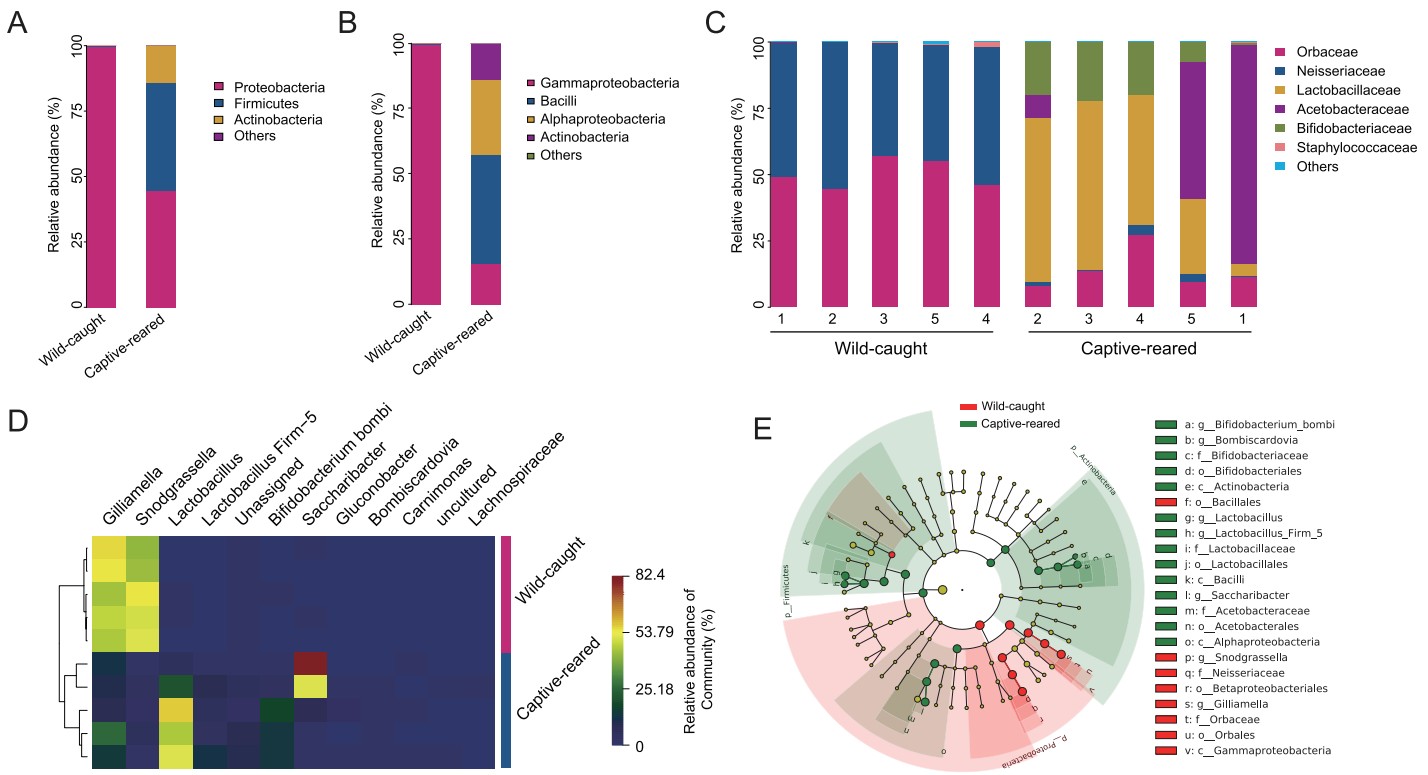

**Figure 3** **Effects of captive-rearing on the gut microbiota composition of *B. lantschouensis*.** (A) The composition of gut microbiota at the phylum level in the wild-caught group and the captive-reared group of *B. lantschouensis*. (B) Composition profiles of gut microbiota at the class level in wild-caught and captive-reared *B. lantschouensis*. (C) The percentage of sequences annotated at the family level in wild-caught and captive-reared *B. lantschouensis*. (D) Hierarchical cluster heat map representation of the top 8 most abundant genera and others in wild-caught and captive-reared *B. lantschouensis*. (E) LEfSe evolutionary branch maps of differences in the gut microbial community between wild-caught and captive-reared *B. lantschouensis*.

gut microbiota in the captive-reared group showed a marked increase in the abundance of Firmicutes, Actinobacteria, and a decrease of Proteobacteria (Fig. 3A). At the class level, the abundance of Gammaproteobacteria (Proteobacteria) was almost 100% in the wild-caught group. In the captive-reared group, Bacilli (Firmicutes), Alphaproteobacteria (Proteobacteria), and Actinobacteria (Actinobacteria) were significantly increased from almost zero to 41.45%, 28.62%, and 14.05%, respectively (Fig. 3B). At the family level, Orbaceae (Orbales) and Neisseriaceae (Betaproteobacteriales) were found to be almost 100% of the wild-caught bumblebee gut microbiota, while in captive-reared bumblebees the dominant composition was Lactobacillaceae (Lactobacillales) and Acetobacteraceae (Acetobacterales), both of which rapidly increased to 41.44% and 28.62%, respectively (Fig. 3C). At the genus level, the most abundant genera (top 12) were displayed in a hierarchical cluster analysis. The abundance of *Lactobacillus* (Lactobacillaceae), Firm-5 (Lactobacillaceae), *Bifidobacterium bombi* (Bifidobacteriaceae), and *Bombiscardovia* (Bifidobacteriaceae) were all significantly increased in the captive-reared group, while almost completely undetected in the wild-caught group. *Gilliamella* (Orbaceae) and *Snodgrassella* (Neisseriaceae) were significantly reduced in the captive-reared group (Fig. 3D). The following bacteria also appeared in the captive-reared group: *Phaseolibacter*

**Table 2 The abundance of sequences annotated at the genera level in the wild-caught group and the captive-reared group of *B. lantschouensis*.**

| Genus | Abundance of sequences | |
|---|---|---|
| | Captive-reared | Wild-caught |
| Gilliamella | 8,627 | 31,054 |
| Snodgrassella | 1,091 | 30,055 |
| Lactobacillus | 22,180 | 16 |
| Saccharibacter | 17,345 | 15 |
| Bifidobacterium bombi/commune/bohemicum | 5,945 | 1 |
| Lactobacillus Firm-5 | 3,313 | 0 |
| Unassigned | 2,650 | 122 |
| Gluconobacter | 261 | 0 |
| Staphylococcus | 0 | 227 |
| Bombiscardovia | 79 | 0 |
| Carnimonas | 12 | 0 |
| Phaseolibacter | 0 | 9 |
| Acinetobacter | 0 | 4 |
| *Lactobacillus kunkeei* | 0 | 3 |

(Enterobacteriaceae), *Acinetobacter* (Moraxellaceae), and *Lactobacillus kunkeei* (Lactobacillaceae; Table 2). In addition, a linear discriminant analysis effect size (LEfSe) was performed and it identified that the order Bifidobacteriales, the family Lactobacillaceae, and the class Actinobacteria were the most important taxa contributing to gut microbiota differences between captive-reared and wild-caught bumblebees (Fig. 3E).

In order to find the possible effects of feeding on host bacteria after capture, a functional prediction analysis was performed using Gene Ontology (GO), Kyoto Encyclopedia of Genes and Genomes (KEGG), and MetaCyc. The results showed that a total of 149 pathways were annotated by the KEGG database. With the heatmap analysis, most of the captive-reared bees exhibited an increase in pathways with an adjusted $P < 0.05$. Five pathways were found to show a decrease. A total of 342 metabolic function pathways were annotated by the MetaCyc database; 123 pathways had significantly different rates of occurrence ($P < 0.05$) between the two groups of bumblebees. The top 30 of these pathways were shown using the expanded histogram with 95% confidence intervals (Fig. S2). These results suggest that physiological and metabolic processes of gut bacterial communities in many hosts may be significantly affected by captive feeding and social environments.

## DISCUSSION

At present, the gut microbiota of more than 20 bumblebee species have been studied, especially in some species of concern such as *B. terrestris*, *B. pascuorum*, *B. ignitus, etc.* Compared with honeybees, rodents, and humans, the gut microbes in bumblebees generally exhibit a simpler community composition (*Wang et al., 2020*). In this study,

many typical gut bacteria of the bee population were detected in *B. lantschouensis*, showing that there is a typical composition in the gut of common bumblebee species (*Hammer et al., 2021*). *Gammaproteobacteria, Bacilli, Alphaproteobacteria*, and *Actinobacteria* were found to be plentiful in *B. lantschouensis*, which is consistent with previous studies of this species and other native species in China (*Koch & Schmid-Hempel, 2011*). Among these, Gammaproteobacteria were the largest proportion of bacteria in the gut of *B. lantschouensis*, which aligns with the findings of a study of their relative *B. terrestris* (*Bosmans et al., 2018*). This may suggest that there is a direct correlation between the proximity of relatives and the composition of the gut microbiota during their coevolution. However, an unusual composition of gut bacteria was also observed in this study. For example, *Bacteroidetes* was a small proportion of the bacterial community, but typically shows significant expression in the gut of other bumblebees (*Koch & Schmid-Hempel, 2011*). *Bohemicum* was also infrequently observed in this study, accounting for only 4.83% of observed microbes. *Bohemicum* may be associated with infection by pathogenic microorganisms and consumes a large number of carbohydrates (*Killer et al., 2009*; *Klocek et al., 2023*). Other bacteria in small proportions, but with special functions, also appeared in *B. lantschouensis*, including: *Staphylococcus, Phaseolibacter, Acinetobacter*, and *Lactobacillus kunkeei*. The unusual composition of bacterial species may be due to species specificity (*Wu et al., 2022*) and diverse food sources.

Artificial feeding can cause major changes in the diet and living conditions of wild animals (*Castaños et al., 2023*). Multiple internal and external conditional signals are induced and exert an influence on the gut microbiota of hosts. The abundance, composition, and even the species of gut bacteria are changed by the rearing conditions of animals such as geckos, birds, and bees (*Sun et al., 2022*). In this study, captive-reared bees showed significant differences in the species and abundance of the bacterial population. Among the decreased populations, *Snodgrassella* (Neisseriaceae) was the most obvious. These bacteria function to supplement bumblebee immunity and productivity in laboratory and field environments (*Cornet et al., 2022*). The *Gilliamella* (Orbaceae) population was also decreased. These bacteria are capable of degrading pollen walls by producing pectinase and are involved in carbohydrate and glycerophosphatide metabolic pathways in their hosts (*Ludvigsen et al., 2018*; *Zhang et al., 2021*), which was confirmed through KEGG analysis, including D−alanine, D−glutamine, and D−glutamate metabolism. The decrease of these two bacteria in general may be related to decreased resistance and decreased energy metabolism, but their remodeling may also be due to ecological changes or the acquisition of new symbiotic organisms as functional substitutes. This warrants further study, but it is clear that these types of bacteria are easily affected by the captive environment, which is consistent with the results of the functional prediction analysis conducted for this study. In the commercial rearing of bumblebees, the addition of probiotic bacterial strains is also very important. Many commercial probiotics are not native to the bee gut but can enhance the health of individual bumblebees and bee colonies as a food additive (*Kim et al., 2024*). In these experiments bumblebees were fed with nothing other than sterilized pollen and sugar water. This might have led to differences in the gut microbiota between the bees in this study and those in typical commercial-rearing
environments (*Ricigliano, Williams & Oliver, 2022*), and it might also provide positive support for the addition of commercial probiotics in rearing.

The abundance of *Lactobacillus* (Lactobacillaceae), followed by *Bifidobacterium* (Bifidobacteriaceae) and *Saccharibacter* (Acetobacteraceae) was increased significantly after rearing. Among *Lactobacillus*, Firm-4 and Firm-5 were most significantly altered. Both are representative species in the bumblebee gut and mainly involved in amino acid metabolic pathways (*Heo et al., 2020*). They usually show antagonistic effects with *Gilliamella* populations and are involved in the enrichment of carbohydrate transport and metabolism (*Su et al., 2022*). This trend reappeared in the overall results, which may reflect common features of *B. lantschouensis* and this study's bumblebees during rearing conditions. *Bifidobacterium* also increased when rearing *B. lantschouensis* like other bees (*Lugli et al., 2022*). Samples were directly collected from adult individuals without standardizing the age of the population, which may affect the abundance of parasitic species in the gut microbiome (*Whitehorn et al., 2011*). These *Bifidobacterium* populations were reported to affect host health *via* regulating host amino acid metabolism (*Satti et al., 2021*). *Bifidobacterium* growth requires carbohydrates as the main nutrients, as shown by the functional prediction analysis in this study. Unlike in other captive-reared bumblebees, *Saccharibacter* showed a significant increase in the results of this study, although it was not the most common bacterium. This bacterium is associated with flower-related microorganisms (*Smith & Newton, 2020*), which may reflect the sensitivity of *B. lantschouensis* to captive-rearing and the difference in genetic background between *B. lantschouensis* and other species. Although this study adopted a laboratory-rearing mode (with a constant indoor temperature of 28 °C), the influence of temperature changes was not taken into account. The temperature was a consistent 28 °C during collection on Xinglong Mountain, but there was a considerable temperature difference between day and night. Such temperature differences might also have an impact on food intake and the gut microbiota (*Palmer-Young et al., 2023*).

A few groups of pathogenic microorganisms were discovered in the field, including *Phaseolibacter*, *Microsporidia*, and *Acinetobacter*, which might be related to the environment. After captive rearing, these wild pathogenic bacteria vanished. Only workers were selected for captive rearing, while the queen and other social factors existing in the wild environment were not considered. The absence of a queen and other social factors might also have an impact on the species of pathogenic microorganisms present (*Bosmans et al., 2018*). It may be that the increased abundance of *Lactobacillus*-related populations prevented the persistent reproduction of pathogenic microorganisms or that the controlled conditions and adequate nutrition contributed to enhanced immunity of the bumblebee host, allowing the diversity of the gut microbiota to maintain equilibrium and suppressing the survival of pathogens (*Alvarez-Perez et al., 2021*). A related example has been reported in a previous study of the common trypanosome parasite *Crithidia bombi* (*Wolmuth-Gordon, Sharmin & Brown, 2023*).

The present study showed that gut bacteria diversity changed in captive-reared *B. lantschouensis* and suggested that wild workers may have increased populations of bacteria involved in metabolism, glycolysis, and the pentose phosphate pathway. This

change in gut bacteria diversity may induce functional changes in the host's carbohydrate metabolism and secondary metabolism such as bile acid biosynthesis. Notably, a rare *Saccharibacter* was found to be increased in the captive-reared *B. lantschouensis*, and pathogenic microorganisms disappeared in captive-rearing conditions. These pathogenic microorganisms were also rare in the gut of wild bees. This study has some limitations. For example, future studies should include the food source of wild bumblebees, as this may have an effect on the intestinal flora diversity in wild-caught *B. lantschouensis*. Also, the classification of ASV is not specific to species and is limited by the database; further study using metagenomics may be beneficial. The exact age of the captured worker bees could not be determined in this study. The lack of a natural nest environment in artificial lab conditions might play a role in these bees maintaining their major symbionts (*Su et al., 2021*).

The role of these influencing factors in the gut microbiota cannot be eliminated. A study by *Weinhold, Grüner & Keller (2024)* discovered that changes in rearing conditions affect the gut microbiota of *B. terretstris* bumblebees. The differences in the experimental scheme and genetic species may explain the differences in results between the present study and *Weinhold, Grüner & Keller (2024)*. The comparison in the present study between captive-reared and wild-caught populations aimed to provide a simple and natural example for the future commercialization of local species and to understand the overall changes in the gut microbiota from artificial feeding. These observed changes are supported by those seen in other types of bumblebees (*Roy et al., 2023*), proving that the conclusions of the present study are credible, despite study limitations. Future studies may want to study the flora of domesticated bumblebees released back into the wild to understand the relationship between the gut microbiota of *B. lantschouensis* and their living environments.

## ACKNOWLEDGEMENTS

We wish to thank Liyuan Yao and Zhibo Hou for their assistance in this experiment. We are also greatful to Prof. Xianhui Wang for his caring and continued support.

### Funding

This work was supported by the National Natural Science Foundation of China (Nos. 32160127 and 32360670), and Higher School Scientific Research Project of the Northwest normal univercity (No. KYZZS2025167). The funders had no role in study design, data collection and analysis, decision to publish, or preparation of the manuscript.

### Grant Disclosures

The following grant information was disclosed by the authors:
National Natural Science Foundation of China: 32160127 and 32360670.
Higher School Scientific Research Project of the Northwest normal univercity: KYZZS2025167.

## Competing Interests

The authors declare that they have no competing interests.

## Author Contributions

- Feng Zhou conceived and designed the experiments, performed the experiments, authored or reviewed drafts of the article, and approved the final draft.
- Shuning Sun conceived and designed the experiments, analyzed the data, prepared figures and/or tables, authored or reviewed drafts of the article, and approved the final draft.
- Xinge Song analyzed the data, prepared figures and/or tables, and approved the final draft.
- Yuying Zhang analyzed the data, prepared figures and/or tables, and approved the final draft.
- Zhuanxia Li analyzed the data, prepared figures and/or tables, and approved the final draft.
- Jiani Chen analyzed the data, prepared figures and/or tables, and approved the final draft.

## Data Availability

The raw data are available in the Supplemental Files.

The data is also available at NCBI: PRJNA1120781.

## Supplemental Information

Supplemental information for this article can be found online at http://dx.doi.org/10.7717/peerj.18964#supplemental-information.

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
