# Peer review of "Captive-rearing changes the gut microbiota of the bumblebee Bombus lantschouensis native to China"

_PeerJ, doi:10.7717/peerj.18964_

## Round 0.1 · original submission · Major Revisions

Some significant concerns about the study need to be addressed. In addition, the manuscript requires extensive proofreading and editing for standard scientific English.

Reviewer 1 ·

Basic reporting

- [ ] **Clear and unambiguous, professional English used throughout**. The manuscript requires extensive proofreading and editing for standard scientific English. Grammatical issues appear through the texts and unfortunately these problems often make the intent of statements unclear. Some are particularly sloppy, such as mis-formatting of species names.
- [ ] **Literature references, sufficient field background/context provided**. Several references are used inappropriately. For example citations on lines 64-65 refer to studies of hoverflies and butterflies, when he text implies this is true of bumble bees. The citation to Yost et al. 2023 appears to support the opposite conclusion tot he point being made in the text. In contrast, some obvious background citations are missing from the Introduction, such as to the influential work on bee microbiota by Waldan Kwong and Nancy Moran.
- [ ] **Professional article structure, figures, tables.** Some text would be better in different sections. The paragraph at the end of the Introduction (lines 93-101) goes into too much detail summarizing the Results. The last paragraph of the Intro (lines 108-115) seems more appropriate in Methods. The second paragraph of the Discussion (lines 288-306) seems like Introduction or Methods text. The figures also need work. The first two panels in Fig 1 add no value. Figures 1 and 2 are missing panel lettering. Fig 3 is missing axis labels to identify the metrics. Why does Fig 4B appear as a pit chart when most of the other panels showing relative abundance are (preferably) bar charts? Fig 4, panels D and E seem redundant.
- [ ] **Raw data shared.** While the raw data were shared with reviewers, the manuscript states that data will only be available by contacting the corresponding author. This is inadequate and appears to fall short of the journal's stated policy that data must appear in a public repository.
- [x] **Self-contained with relevant results to hypotheses.**

Experimental design

- [x] **Original primary research within Aims and Scope of the journal.**
- [ ] **Research question well defined, relevant & meaningful. It is stated how research fills an identified knowledge gap**. This study is exploratory. I don't really object to that fact, but it does not test a hypothesis and does not fill a particularly critical gap in the field.
- [ ] **Rigorous investigation performed to a high technical & ethical standard.** An important issue of experimental design is that individual workers are pooled (if I read the Methods on lines 131-133 correctly). Why? This will reduce inter-sample variance. I know from experience that sufficient DNA can be obtained from the gut of one bumble bee to allow 16S microbiome sequencing.
- [ ] **Methods described with sufficient detail & information to replicate.** The Methods have several gaps. One critical omission from the Methods however is a clear statement about sampling: were all individuals collected on the same rose plants? This directly impacts the discussion about variation in wild-caught individuals. What method was used for the phylogenetic inference in Fig 4F? Many software programs are not cited or cited incorrectly. For example, Siddiqui et al 2022 is not the original source of those methods.

Validity of the findings

- [x] **Impact and novelty not assessed. Meaningful replication encouraged where rationale & benefit to literature is clearly stated.**
- [ ] **All underlying data have been provided; they are robust, statistically sound, & controlled.** One important limitation of this study design goes unaddressed. The investigators sequenced microbiota from wild-caught workers and from workers held captive for 20 days in what appears to be a common arena. Thus the social environment experienced by the individuals is a confounding factor, not simply "wild" vs. captive. A better approach would be to rear artificial colonies and compare those kept fully captive from those allowed to forage openly, therefore controlling for the social environment. The Methods and Results section also disagree on the statistical tests being used. ANOVA is mentioned in the Methods; Adonis (that is, PERMANOVA) in the Results. Other analyses are simply not described in sufficient detail, such as the relative abundance ("indicator") analysis in Figure 5. (Is that based on loads in the PCoA?) -- Moreover the graphic design in Figure 5 is terrible!
- [ ] **Conclusions are well stated, linked to original research question & limited to supporting results.** Many of the conclusions are highly speculative. There is no evidence that many of the more diverse ASVs identified as pathogenic, as the text implies.

Additional comments

Host-microbiome interactions are a topic of broad interest, as more and more aspects of host biology are revealed to take influences from gut microbiota. Bumble bees in particular present an interesting case study in this area, because of their social lifestyle, ecological and agricultural importance. The manuscript by Zhou et al. presents the characterization of microbiota from wild-caught and captive-reared workers of *Bombus lantschouensis*. This topic seems relevant to the community of bumblebee microbiome research and perhaps to a wider audience concerned with microbiome-host interactions. Unfortunately, the manuscript suffers from several problems, that in my estimation preclude publication at this time.

·

Basic reporting

Major criticism refers to the English of this manuscript. Many sentences are grammatically wrong or make little sense at all. It is highly advised that English proof-reading service will be applied. This becomes immanent when reading the abstract (line 33, 37, 38, 39, 40, 41, 43) or the title of this article which probably translates into: “Indoor-rearing changes the gut microbiota of the bumblebee Bombus lantschouensis native to China”. Other example are line 39: “The species of genus Gilliamella and Snodgrassella were the advantage bacterium.” Line 42: “…were dramatically increased after indoor reared and server as the dominant bacteria.” Line 70: “For difference inherent attributes, gut microbiota often showed asynchronous in dynamically changing in community.” I am not sure if a bad machine translation was the problem here, but it should be cross checked by a human being. This created also some nonsense phrases like “significant pollinators” or “ecological pollination”.
Despite the English, the content of the introduction is acceptable. Discussion can be improved (see details below). Figures with major findings are quite clear, but not all figures seem necessary (see details below).
As far I can tell the raw data has not been uploaded into any repository like GenBank. The data availability statement mentions that it is “available from the corresponding author on reasonable request”.

Experimental design

Regarding the experimental design descriptions are not fully clear. Initially it reads that “60 wild workers” had been sampled (line 118), which were divided in to two groups (2x30 samples). Then it reads that “Gut samples from 30 bees of each treatment group were divided into three biological replicates including 10 gut samples” which to my understanding means that 10 guts were pooled per group so that only 3 biological replicates remained (2x3 samples). This seems to be confirmed in line 139, which mentions that “DNA samples with three replicates each group were used for PCR amplification.” However, results mention that “A total of ten DNA samples were amplificated”, which seems to be finally confirmed in Table S1 and Fig. 3 and Fig.4. So the entire analysis is based on two groups with each contain only 5 samples (2x5 samples), right? What remains unclear is still if these resemble pooled samples (30 samples from which each 6 guts were pooled into 5 pooled samples?), or if samples of minor quality have just been dropped out during analysis? This needs to be clarified in the text. Also, the reason for reducing replicate numbers so drastically although 60 individuals were available is not given.

Though this manuscript describes basically only a single experiment, the description is not complete. Names and references for the used primers are not provided. Type of polymerase is not provided (proof- or non-proof reading?). Details about amplicon library construction are not provided. Filtering criteria from line 180 and Table S1 are not provided. Also, for a possible interpretation of the main findings the indoor rearing conditions need to be precisely described. What are the “standard laboratory conditions” of line 126? Temperature, size of colony, mortality and performance of colony indoors, sterility or non-sterility of provided food are all important factors to understand why indoor microbiota might have changed.

Validity of the findings

Still, despite the very low replication provided here, the investigation of microbiome decay of wild bumble bees following artificial rearing is in general interesting. Though broader implications are quite limited, the presentation of the main results is largely okay. These graphs are quite good to convey the message. I think Fig. 3B-E and Fig. 4A-E can remain as they are, as they show in a clear manner how community composition differs in both groups. But Fig 1 + 2 provide only little extra information, especially since the composition data of both groups had been pooled in Fig 2B. On the other hand Fig 4F,G and Fig 5 seem unnecessary as they bring not much new information and are largely redundant to the previous findings. Functional predictions of Fig. 6 seem unnecessary and misleading. Those analysis are anyway highly speculative and rarely supported by independent experiments. But here they give the impression of gain of functions in indoor conditions (line 262 “higher abundance in 30 pathways”). I think this is misleading for a degenerated indoor microbiome which lacks the two most important symbionts.

In general, the discussion focusses little on the main findings itself but refers more to technical issues of the applied method or broader composition of bumble bee microbiota. I think the paragraph about technical aspects (line 288- 306) could be removed entirely. The focus should me more on microbiome decay and changes in indoor rearing and its implications for commercial bumble bee suppliers. The authors do not discuss possible explanations why the microbiome has changed in their small in indoor rearing study. One important factor could be change of temperatures, which are sadly not provided for their rearing conditions. Another issue could be social isolation or disturbed sociality (only worker but no queen?). Aging differences of 20 days can be important as well. How old do these bumblebees usually get? Are these maybe indications of very old dying individuals? Also, the lack of their natural nest environment in the artificial lab conditions might play a role to maintain their major symbionts? All this has not been mentioned nor discussed properly. Even other literature with similar or contradicting patterns has not been much discussed. There are similar studies that show even the same pattern just with the opposite direction (shifting indoor to outdoor), where the bumble bee microbiota shows an increase of lactic acid bacteria and a decrease of Snodgrassella when exposed to outdoor environments. This would be more meaningful than the implementation of mammalian examples.
I think a focus on the actual finding (change of microbiome indoors) and a proper discussion of this (how and why it changes and what the consequences are) would improve this manuscript.

Additional comments

The authors also briefly mention COI sequencing, without providing further details (primer?).
Abstract and Discussion mention also the occurrence of potential pathogens, which have not been describe in the results at all.

---

## Round 0.2 · accepted · Accept

The authors have addressed all the concerns of the reviewers